

# The effects of gamelike features and test location on cognitive test performance and participant enjoyment

Jim Lumsden[1,2], Andy Skinner[1,2], Andy T. Woods[2], Natalia S. Lawrence[3] and Marcus Munafò[1,2]

[1] MRC Integrative Epidemiology Unit (IEU), University of Bristol, Bristol, United Kingdom
[2] School of Experimental Psychology, University of Bristol, Bristol, United Kingdom
[3] School of Psychology, College of Life and Environmental Sciences, University of Exeter, Exeter, United Kingdom

## ABSTRACT

Computerised cognitive assessments are a vital tool in the behavioural sciences, but participants often view them as effortful and unengaging. One potential solution is to add gamelike elements to these tasks in order to make them more intrinsically enjoyable, and some researchers have posited that a more engaging task might produce higher quality data. This assumption, however, remains largely untested. We investigated the effects of gamelike features and test location on the data and enjoyment ratings from a simple cognitive task. We tested three gamified variants of the Go-No-Go task, delivered both in the laboratory and online. In the first version of the task participants were rewarded with points for performing optimally. The second version of the task was framed as a cowboy shootout. The third version was a standard Go-No-Go task, used as a control condition. We compared reaction time, accuracy and subjective measures of enjoyment and engagement between task variants and study location. We found points to be a highly suitable game mechanic for gamified cognitive testing because they did not disrupt the validity of the data collected but increased participant enjoyment. However, we found no evidence that gamelike features could increase engagement to the point where participant performance improved. We also found that while participants enjoyed the cowboy themed task, the difficulty of categorising the gamelike stimuli adversely affected participant performance, increasing No-Go error rates by 28% compared to the non-game control. Responses collected online vs. in the laboratory had slightly longer reaction times but were otherwise very similar, supporting other findings that online crowdsourcing is an acceptable method of data collection for this type of research.

Corresponding author
Jim Lumsden,
jim.lumsden@bristol.ac.uk

## INTRODUCTION

Cognitive tasks are a common tool in psychological research, but are often viewed as effortful and unengaging by participants (*D'Angiulli & LeBeau*, *2002*; *Healy et al.*, *2004*). If participants are bored by an assessment, then their lack of motivation may have negative effects on data quality, adding noise and leading to suboptimal performance (*DeRight & Jorgensen*, *2014*). One potential solution is to add gamelike elements to these tasks in

order to make them more intrinsically enjoyable (e.g., *McPherson & Burns*, *2008*; *Prins et al.*, *2011*). By using game mechanics which incentivise maximal performance, participants' goals might be adjusted from 'completing the experiment as quickly as possible' to 'trying to succeed at the game,' thus producing better data (*Hawkins et al.*, *2013*). For example, a cognitive task with a points mechanic might incentivise rapid responding and consistently high accuracy by awarding points in relation to these measures.

However, the assumption that a more engaging task will provide better data is largely untested. Some studies have reported a positive effect of game mechanics on participant performance (*Dovis et al.*, *2011*; *Ninaus et al.*, *2015*), although most have shown mixed results (e.g., *Katz et al.* (*2014*); see *Lumsden et al.* (*in press*) for a review). There is some evidence that gamelike tests do *not* improve performance, and may in fact worsen it (*Hawkins et al.*, *2013*), potentially by introducing new task demands. In contrast, many studies have shown that gamelike experiments are more motivating and less boring (*Dörrenbächer et al.*, *2014*; J Lumsden, A Attwood & M Munafo, 2013, unpublished data; *Mcpherson & Burns*, *2007*; *Tong & Chignell*, *2014*).

*Miranda & Palmer* (*2014*) investigated the effects of two different game mechanics, sounds and points. They found that sound effects slowed reaction times, but points did not, showing that some game elements may have more impact on task performance than others. Further systematic research is required to understand how specific gamelike features impact the quality of data gathered from, and participant ratings of, cognitive tasks.

Another recent development in cognitive research is the deployment of cognitive tasks on online platforms. One of the key enablers of online research is Mechanical Turk (MTurk; www.mturk.com), an Amazon-based 'work marketplace' which allows users to sign up, complete small online tasks and receive reimbursement for their time. While MTurk is often used for non-research purposes, it has grown popular in the behavioural sciences because it enables the testing of large numbers of people in a very short time. However, studies investigating the comparability of data from laboratory and online testing versions of tasks have again reported mixed findings (*Crump, McDonnell & Gureckis*, *2013*; *Schreiner, Reiss & Schweizer*, *2014*). These differences may arise from a number of factors, including: differences in the population sampled (with online participants tending to be older than those recruited through traditional methods), differences in hardware used to run a given study, the suitability of the remote environment for concentration and reduced motivation due to lack of experimenter presence (see *Woods et al.*, *2015*). Interestingly, *Hawkins et al.* (*2013*) also found that participants' self-reported enjoyment and engagement was much lower when the task was online, but it was unclear why this was the case.

In this study we aimed to investigate the effects of gamelike features and the effects of test location on the data collected by and participant enjoyment of a simple cognitive task. We used three variations of the Go-No-Go task (GNG), delivered both in the laboratory and online using MTurk: one variant where participants were rewarded with points for performing optimally, one where the task was framed as a cowboy shootout (game theme), and a standard GNG task as a control condition. Both the laboratory and online arms of the study used Xperiment, a web-based platform for psychological experiments (www.xperiment.mobi), as a delivery method.

## METHODS

### Design and overview

The aim of this study was to compare three versions of GNG-task, each with different gamelike features (non-game, points, theme) across two different testing sites (laboratory and online). We used a between-subjects design, with reaction times (RT) on Go trials, Go trial accuracy, No-Go trial accuracy and subjective ratings as the dependent variables of interest. We did not expect to see a difference in median RTs or mean No-Go accuracy between any of the GNG variations. We had no expectations regarding differences in Go accuracy between the task variants. Finally, based on effects found previously, we did anticipate that participants would rate both the theme variant and points variant favourably compared to the non-game control. We also expected all ratings to be lower on average in the online.

### Participants

Participants who were tested in the laboratory were staff and students recruited through existing email lists and poster advertisements around the University of Bristol. They received either course credit or £3 in compensation for their time. Participants who were tested online signed up to the study through MTurk; they received payment of $1.50. We required that participants were older than 18 years of age, did not have a diagnosis of ADHD and were not colour blind.

Once enrolled, participants were randomly assigned to one of the three task variants. Since testing site (laboratory or online) was determined by the participant's method of signup, the groups were not matched. The laboratory condition included 28 participants in each task variant and the online condition included roughly 72 participants in each task variant. Precise allocation of equal numbers of participants to each task variant could not be achieved online due to multiple concurrent signups to the experiment-platform.

### Materials

#### Online and laboratory platforms

In order to eliminate task differences caused by variations in delivery platform, we used Xperiment to host both the lab and the online version of the task. Xperiment is an online experimental platform which has been shown to collect comparable data to other, offline, test software (*Knoeferle et al.*, *2015*; *Michel et al.*, *2015*). Laboratory participants were seated in a computer cubical while they completed the task and the questionnaire via the internet. They used a PC with a mouse and keyboard to complete the task. MTurk participants accessed exactly the same experimental software, but via their own PC or laptop.

#### Go/No-Go task

The Go/No-Go task (GNG) is a measure of response inhibition (the ability to stop or withhold a motor response), which is a key feature of executive control (*Verbruggen & Logan*, *2008*). The main cognitive tasks that are used to assess response inhibition include: the stop-signal task, which imposes a delay between a stimulus and a stop signal, thus placing demands on 'action cancellation' (inhibition of an initiated response); and the GNG task, which is a qualitatively different 'action restraint' task (*Schachar et al.*, *2007*). It

comprises a reaction-time task with a set of fixed no-action stimuli. It measures inhibitory control by repeatedly presenting stimuli to which the participant must respond rapidly, while occasionally presenting stimuli to which the participant must avoid responding.

We developed our own GNG task for use on the Xperiment platform, based on the tasks used by *Benikos, Johnstone & Roodenrys* (*2013*) and *Bowley et al.* (*2013*), but with custom features for each variant. Each trial began with a fixation cross displayed in the middle of the screen, 500 ms later a picture appeared in the centre of the screen and remained for 600 ms. On Go trials the participant had to respond to the stimuli as fast as they could by pressing the spacebar within this 600 ms window. In No-Go trials (signalled by the image content) they simply had to withhold their response. Each trial was followed by a variable inter-trial-interval (ITI) of 500–1,000 ms. If the participant responded incorrectly, the ITI was replaced by a feedback screen, failed No-Go trials resulted in a red cross overlaid on the stimuli, while incorrect no-responses were followed by "Too slow" written in red text.

The task consisted of 5 blocks of 60 trials each. Between each block a pause screen was displayed and the participant had to wait for 10 s. Each block contained 5 sub-blocks of 12 trials, and each sub-block consisted of 9 Go trials and 3 No-Go trials, in randomised order. In total, the task contained 75 No-Go trials (25%) and 225 Go trials (75%) and took around 11 min to complete. GNG tasks vary widely in their design, but using 25% No-Go trials is similar to several other studies (*Benikos, Johnstone & Roodenrys*, *2013*; *Kertzman et al.*, *2008*; *Watson & Garvey*, *2013*; *Yechiam et al.*, *2006*).

**Non-game variant:** The non-game control used a stimulus set consisting of a diverse range of 20 everyday objects: 15 green and 5 red. Go trials used the green object, and No-Go trials used the red objects (see Fig. 1). We selected green and red objects to ensure that the non-game variant was as intuitive as the themed variant, as these colours are commonly associated with going and stopping (*Moller, Elliot & Maier*, *2009*). See Fig. S1 for the instructions presented to the participants.

**Points variant:** The points variant was identical in structure to the non-game control, except that a scoring system was added, based on that used in *Miranda & Palmer* (*2014*). The participant's score was displayed in middle of the screen, to the left of the stimuli (see Fig. 1). On each successful Go trial the participant earned points equal to $bonus*(600-RT)/10$. This *bonus* was a multiplier ($2\times, 4\times, 8\times\ldots$) which doubled every 5 trials but was reset to $\times 1$ when the participant made a No-Go error. On a successful inhibition the bonus was not lost, but no points were awarded. This reward/punishment scheme also fits with findings of *Guitart-Masip et al.* (*2012*), who found that subjects were much more successful in learning active choices when rewarded for them, and passive choices when punished. The points awarded in the previous trial were displayed in the centre of the screen during the ITI. The instructions framed the task as a game, see Fig. S2.

**Theme variant:** The theme variant also used the same format as non-game control, except with the addition of a theme designed to provide a narrative framework for the action required by the task (see Fig. S3). The participant was introduced to the task as a shooting game, where they were the sheriff of a small town and a group of criminals had holed up in a saloon and taken hostages. The GNG task proceeded as above but the stimuli were replaced

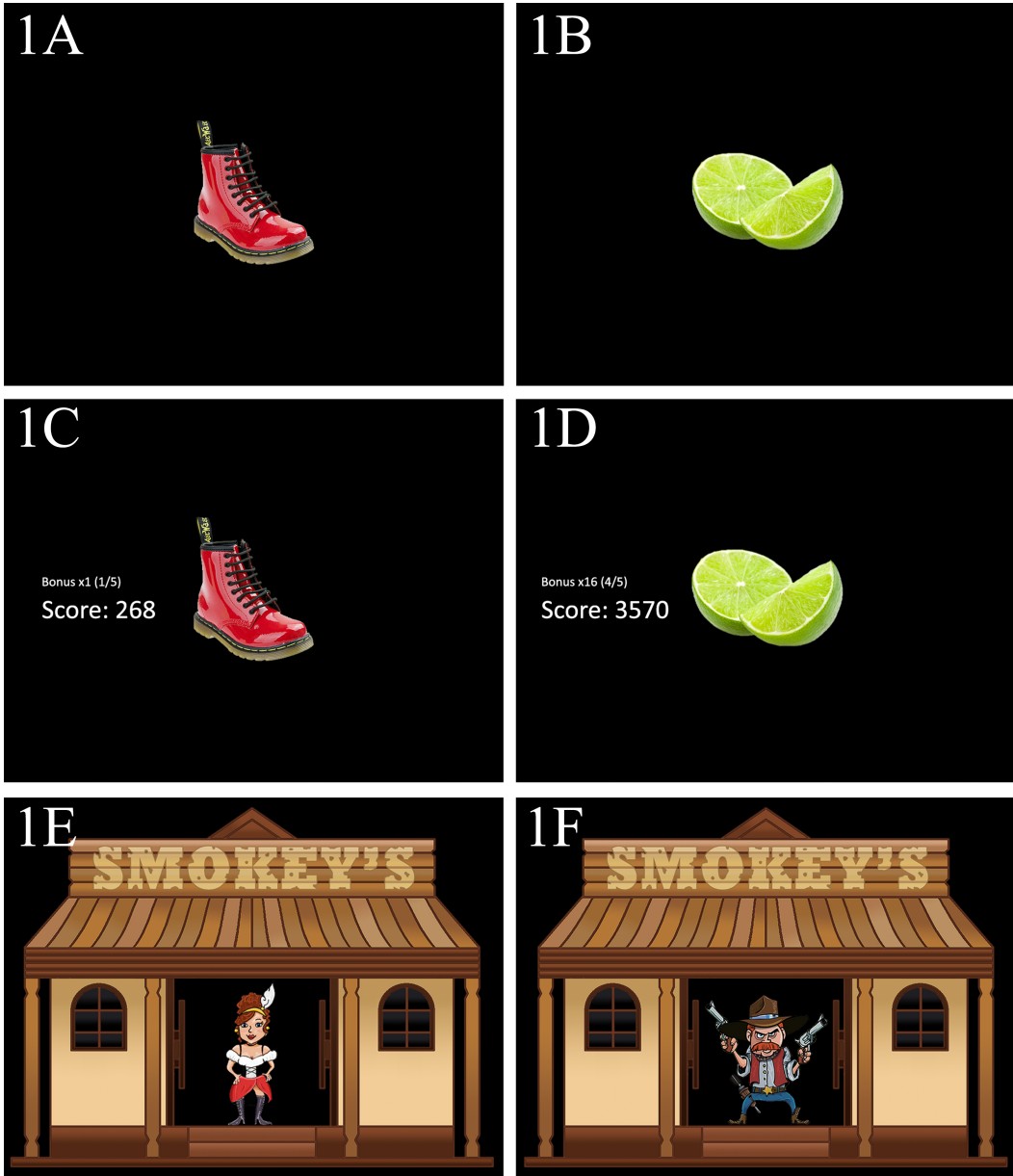

**Figure 1** (A) No-Go trial from the control variant. (B) Go trial from the control variant. (C) No-Go trial from the points variant. (D) Go trial from the points variant. (E) No-Go trial from the theme variant. (F) Go trial from the theme variant.

with cartoon characters; with cowboys as Go targets and innocent civilians as No-Go targets (see Fig. 1). Throughout each block a cartoon saloon graphic remained on the screen, with stimuli appearing in the doorway. When the participant pressed the response key, a blood splat was overlaid onto the current stimuli for the remainder of the trial time. Feedback was presented in the ITI, as in the non-game control. The stimulus set consists of 15 cowboys and five innocent civilians.

### Assessment of enjoyment and engagement

After completing the task participants were given a short questionnaire to assess their opinion of the task. Following assessment approaches by *Hawkins et al.* (*2013*) and *Miranda & Palmer* (*2014*), 11 questions were selected: (1) How enjoyable did you find the task? (2) How frustrating did you find the task? (3) Was it difficult to concentrate for the duration of the task? (4) How well do you think you performed on this task? (5) How mentally stimulating did you find this task to be? (6) How boring did you find the task? (7) How much effort did you put in throughout the task? (8) How repetitive was the task? (9) How willing would you be to take part in the study again? (10) How willing would you be recommend the study to a friend? (11) How intuitive did you find the pictures chosen for stop and for Go? Participants responded using a continuous visual analogue scale (VAS), presented as a horizontal line with a label at either end and no subdivisions. Participants marked a point between these two labels using their mouse. The questionnaire was delivered using the same Xperiment platform that delivered the tasks.

### Procedure

Study sessions lasted approximately 15 min. Each participant took part in only one task variant in order to minimise the duration of the study and prevent fatigue. Participants confirmed that they met the inclusion criteria and provided consent using an online form. We then collected demographic information on the participant's age, sex, ethnicity, level of education and the number of hours they spent playing video games per week. Instructions for the task were then displayed. The GNG task was then delivered, followed by the questionnaire and finally a debrief screen was displayed. Participants were free to withdraw from the study at any point by simply closing the browser window, this would result in no data being saved. The study was pre-registered on the Open Science Framework (https://osf.io/va547/) and ethical approval was obtained from the Faculty of Science Research Ethics Committee at the University of Bristol (22421). The study was conducted according to the revised Declaration of Helsinki (WMA Declaration of Helsinki—Ethical Principles for Medical Research Involving Human Subjects: 2013, October 19).

### Statistical analysis

Since we did not anticipate a difference in RTs or mean No-Go accuracy between task variants, we initially decided not to use a Frequentist approach as it is not ideal for testing equivalences (*Berger & Sellke*, *1987*; *Blackwelder*, *1982*): Bayesian analyses are better suited to this (*Wetzels et al.*, *2009*). However, upon collection and initial exploration of the data it was apparent that large differences did exist and so we updated our statistical plan to include both Frequentist and Bayesian approaches.

### Sample size determination

At the time of study design, no previous study had investigated differences in data produced by gamelike and non-gamelike GNG tasks, and therefore we had no previous effect size on which to base a sample size determination. We selected a sample size for the laboratory condition to provide sufficiently dense distributions to allow for meaningful analysis. For

**Table 1 Interpreting Bayes factors (adapted from *Raftery*, *1995*).**

| Hypothesis 0: The difference between means is 0 | Strength of evidence | Hypothesis 1: The difference between means is between 0 and X |
|---|---|---|
| $.33 \leq BF \leq 1$ | No support either way | $1 \leq BF \leq 3$ |
| $.1 \leq BF \leq .33$ | Positive | $3 \leq BF \leq 10$ |
| $.01 \leq BF \leq .1$ | Strong | $10 \leq BF \leq 100$ |
| $BF < .01$ | Decisive | $BF > 100$ |

the online condition we scaled up our sample-size to take advantage of the larger samples possible with crowdsourcing.

### Reaction time data

Reaction time data were summarised by median Go RTs for each participant. Differences between task variants and testing sites were assessed using box-plots and two-way ANOVAs. Where Frequentist approaches found no evidence of a difference between two means, we used Bayesian $t$-tests to assess the evidence for equality (*Rouder et al.*, *2009*). A Bayesian $t$-test produces a Bayes Factor, which either provides evidence to support one of two hypotheses, or implies the data are insensitive, see Table 1. In our analysis one hypothesis was always "the mean difference is zero" and the other was "the mean difference is not zero." We used the Bayesian $t$-test procedure from the R-Package BayesFactor (http://bayesfactorpcl.r-forge.r-project.org/), with a naïve JZS prior.

### Accuracy data

Accuracy data were handled similarly. We calculated % accuracy scores on Go and No-Go trials for each participant. Differences between task variants and sites were assessed using box-plots and two-way ANOVAs. Where we found no evidence of a difference between two means, we used Bayesian $t$-tests to weigh the evidence for equality.

### Questionnaire data

We assessed differences in participant ratings both visually and using a two-way ANOVA of total score with site and task variant as factors. Total score was computed by averaging the VAS scores from items 1–10 (with items 2,3,6 and 8 reversed) to produce a score out of 100.

## RESULTS

### Characteristics of participants

A total of 304 participants took part in this study, however four participants from the online group were excluded from subsequent analyses because we did not record any responses from them for the duration of the GNG task. A further thirteen participants were excluded from the analysis due to extremely poor Go accuracy rates (more than 4 inter-quartiles ranges away from the median).

Excluding outliers, 287 participants took part: 84 in the laboratory (mean age = 21, SD = 4, 26% male) and 203 online (mean age = 35, SD = 11, 50% male). A chi-square test indicated that the number of male participants in the laboratory site was statistically different

to the online ($X^2$ (1, $N = 287$) $= 14.012, p < .001$). A $t$-test provided evidence for difference in ages between the laboratory group and online ($t(285) = 16.35, p < .001$), with the online participants typically being older. Participants who took part online reported slightly more hours spent playing computer games per week (median = "1–5") than those that took part in the lab (median = "0")—there was evidence that the distributions of responses for both groups differed, with the laboratory group being skewed towards 0 (Mann–Whitney $U = 3,994$, Online $= 203$, Lab $= 84, p < .001$ two-tailed). Online participants also reported higher levels of education (median = "Bachelor's degree") than those in the laboratory (median = "High School Graduate"), and there was evidence that these distributions differed, with 83% of the laboratory group being high school graduates and the online group being a relatively even split between high school graduates and university graduates (Mann–Whitney $U = 5,330$, Online $= 203$, Lab $= 84, p < .001$). However, given that the majority of laboratory participants were undergraduates, they will be more than equally educated within a few years. Ethnicity also differed between sites ($X^2$ (4, $N = 287$) $= 20.456, p < .001$): both groups featuring a high proportion of participants of European ancestry (69% in the laboratory, 85% online), but we saw a higher proportion of East Asian participants in the laboratory sample (14% vs. 4%). Screen resolution in the laboratory was $1,920 \times 1,080$, median screen resolution online was $1,440 \times 900$.

## Go trial data

Data from Go trials in all three variants and on both sites are shown in Figs. 2 and 3 and Tables S1 and S2. A two-way ANOVA of the median Go RTs indicated main effects of both task variant ($F[2, 281] = 174.891, p < .001, \eta^2 = .56$), and site ($F[1, 281] = 24.906, p < .001, \eta^2 = .08$); however, there was no evidence of an interaction ($p = .298$). Go RTs were longer online and were also longest in the theme variant. Post-hoc $t$-tests showed RTs from the theme variant to be longer than the points ($t(190) = 16.316, p < .001, d = 2.37$) and non-game ($t(186) = 16.991, p < .001, d = 2.49$) variants; however, we could not detect a difference between the points and non-game variants ($t(192) = .085, p = .932, d = .01$). We therefore compared the non-game and points variants using a Bayesian $t$-test and found good evidence that Go RTs were equal in the non-game and points variants (Bayes factor $= 0.157$). We also performed exploratory analysis into the effect of task duration on RT, see the Supplemental Information 1 document.

Accuracy followed a similar pattern. A two-way ANOVA found evidence for main effects of both task variant ($F[2, 281] = 72.974, p < .001, \eta^2 = .34$) and site ($F[1, 281] = 15.277, p < .001, \eta^2 = .05$). Again, there was no clear evidence of an interaction ($p = .143$). Go accuracy was generally very high, as expected. However, it was slightly lower online, see Fig. 3. Post-hoc $t$-tests showed that the theme variant had lower accuracy than the points ($t(104.1) = 10.347, p < .001, d = 2.03$) and non-game ($t(115.8) = 9.413, p < .001, d = 1.75$) variants. We could not detect a difference between the points and non-game variants ($t(170.9) = 1.511, p = .133, d = .23$) and a Bayesian $t$-test to compare the points and non-game variants for equality suggested there was insufficient evidence to support either equality or a difference (Bayes factor $= 0.459$). Due to the non-normality of the data, we also used Mann–Whitney $U$ tests to confirm the ANOVA findings, see Table S3.

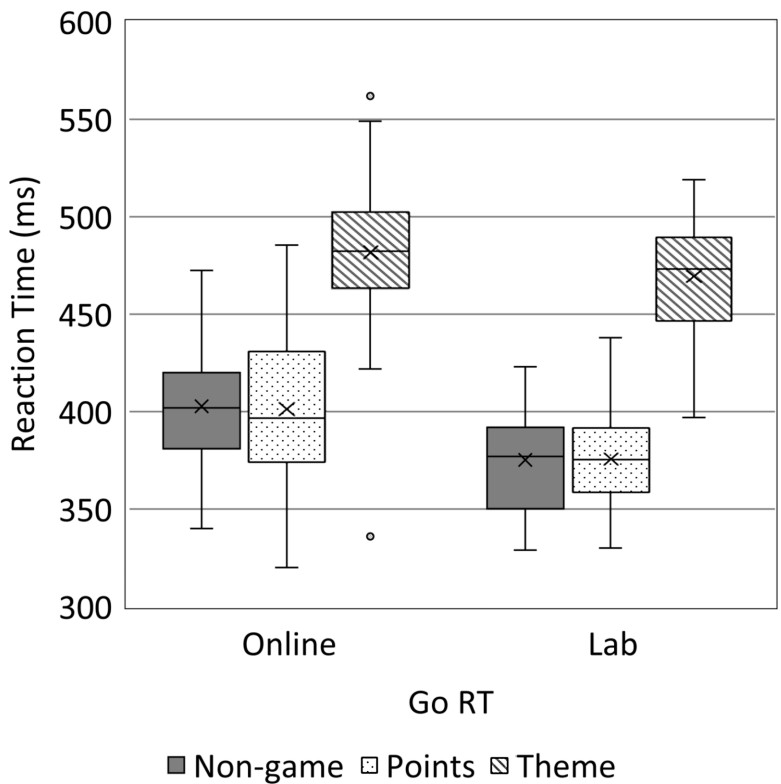

**Figure 2** Box and Whisker plots of Median Go RTs, split by task variant and site.

### No-Go trial data

Data from No-Go trials in all three variants and from both sites are shown in Fig. 3 and Tables S1 and S2. A two-way ANOVA of No-Go accuracy data found evidence of a main effect of task variant ($F[2, 281] = 247.362$, $p < .001$, $\eta^2 = .64$), but no evidence for an effect of site or an interaction ($ps > .393$). No-Go accuracy was much lower in the theme variant than the other two variants, and post-hoc $t$-tests showed that the theme variant was different to the points ($t(106.5) = 18.396$, $p < .001$, $d = 3.57$) and non-game ($t(114.7) = 17.582$, $p < .001$, $d = 3.28$) variants. Again, we could not detect a difference between the points and non-game variants ($t(180.9) = 1.012$, $p = .313$, $d = 0.15$) but a Bayesian $t$-test found good evidence that No-Go accuracy was equivalent in the non-game and points variants (Bayes factor $= 0.253$). Additionally, we performed exploratory analysis into the effect of task duration on No-Go accuracy, see the Supplemental Information 1 document.

We saw ceiling effects in both the points and non-game variants, which resulted in skewed distributions. Due to the non-normality of the data, we used Mann–Whitney $U$ tests to check the results of the post-hoc $t$-tests of Go and No-Go Accuracy between task variants, see Table S3. All Mann–Whitney $U$ tests confirmed the findings of the $t$-tests.

### Subjective questionnaire of enjoyment and engagement

Table S4 shows the mean VAS scores from the engagement questionnaire, by site and task variant. In general, subjective engagement scores were slightly higher online ($t(285) = 2.732$,

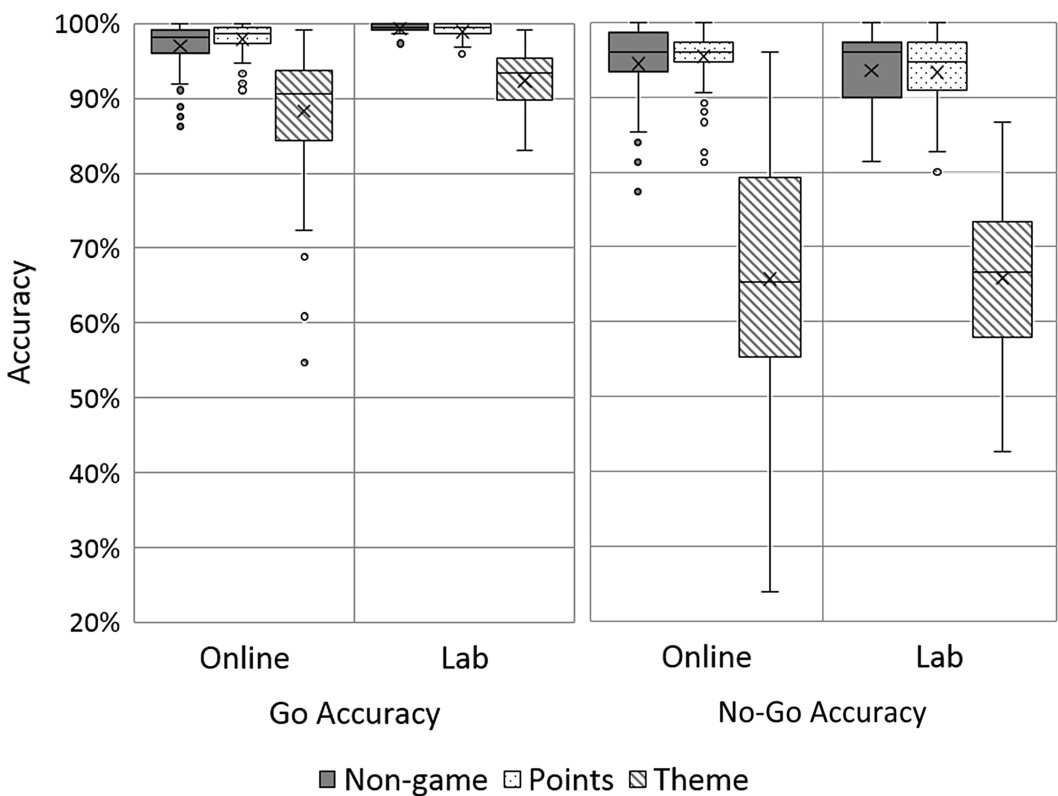

**Figure 3  Box and Whisker plots of Go and No-Go Accuracy, split by task variant and site.**

$p < .001$, $d = 0.32$). Online participants rated all the task variants as more repetitive than those in the laboratory group, yet were much more willing to take part in the study again.

A two-way ANOVA of total-score data found evidence of a main effect of task variant ($F[2, 281] = 3.613$, $p = .028$, $\eta^2 = .03$) and site ($F[1, 281] = 7.423$, $p = .007$, $\eta^2 = .03$). Post-hoc $t$-tests showed that the non-game variant was rated lower than the points variant ($t(192) = 2.986$, $p = .003$, $d = 0.43$), but no other differences were found ($ps > .178$).

We suspected that the heterogeneity in group composition might be driving the difference in total score between the laboratory group and the online group. To assess we performed a two-way ANCOVA of total-score data with site and task variant as factors, and age and sex as covariates. Again we found evidence of a main effect of task variant ($F[2, 279] = 3.070$, $p = .048$, $\eta^2 = .02$), but not for site, sex, age or an interaction ($ps > .084$). This implies that the difference in scores between the two sites was indeed an artefact of age/sex preferences, and that task variant was the primary factor driving a difference in scores, see Fig. 4.

We performed two Bayesian $t$-tests to investigate whether the total scores of either the points and theme variants or the non-game and theme variants could be considered equal, but found that the data supported neither equality or inequality (Bayes factors = 0.322 and 0.372 respectively).

Figure 5 shows the individual item questionnaire scores broken down by task variant alone. The pattern found in the total scores is apparent in the individual questionnaire items also, with the points variant being rated slightly more favourably. The non-game

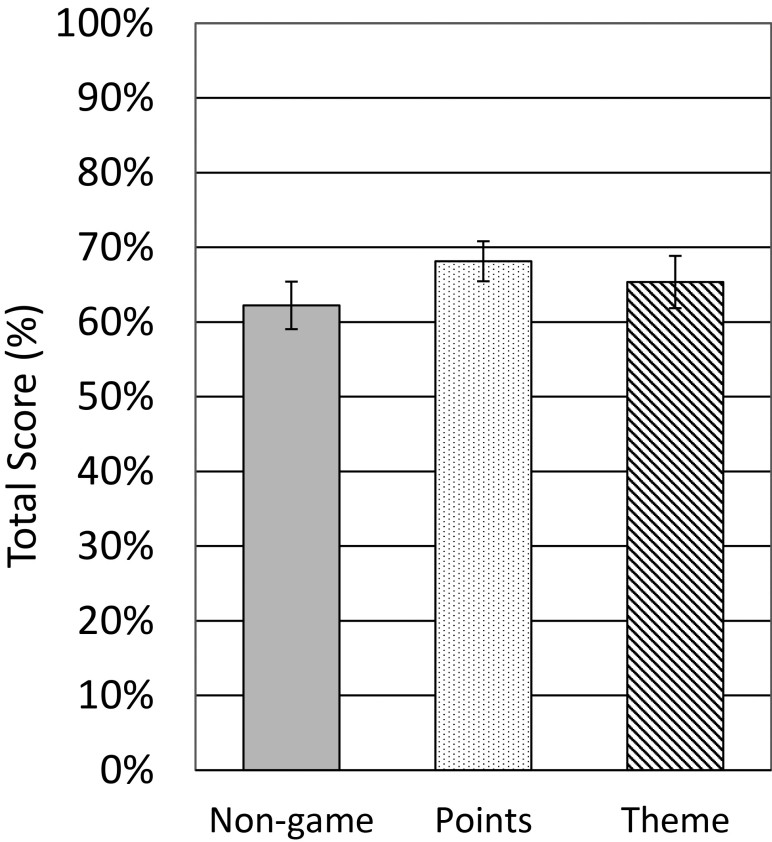

**Figure 4** Mean total scores from the subjective questionnaire, split by variant and adjusted for age and sex.

control was clearly rated as the least enjoyable and stimulating, the most boring and the most frustrating. Participants also reported putting less effort into this variant than others. The theme variant had mixed scores, with participants feeling they performed poorly and finding it very frustrating; however, it does appear that the cowboy stimuli resulted in the task being less repetitive, and on several measures, such as enjoyment, it does not differ from the points variant. Overall, the points variant was best received: these participants were the most willing to recommend the study to a friend, as well as willing to put the most effort into the task. We found no difference between any of the three variants on ratings of "difficulty concentrating" or "intuitive pictures."

# DISCUSSION

## Comparison of task site (online vs. laboratory)

The laboratory group was a fairly unrepresentative sample consisting mainly of young, female undergraduates who volunteered for the experiment. The MTurk group had a much more balanced demographic, with a range of ages, education levels and games experience. Although MTurk users are also a self-selected group, their slightly wider demographic lends some ecological validity to our findings. We also saw some differences between the online and laboratory group in terms of the behavioural data we collected. RTs recorded from MTurk

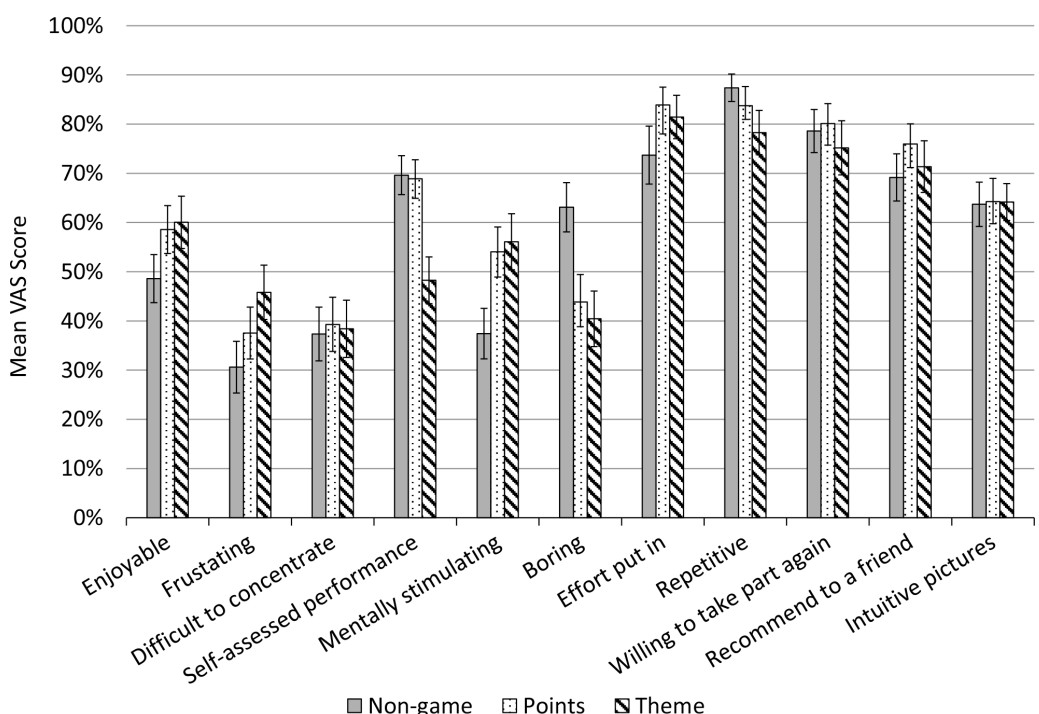

**Figure 5  VAS scores for individual questions on the subjective engagement questionnaire, split by task variant.**

users were ∼25 ms longer on average. Notably, we saw no difference in No-Go accuracy, implying that online participants were just as able to inhibit their responses to stimuli.

There are several possible reasons for the longer RTs online. Lower participant effort may have played a role, potentially resulting from the absence of an experimenter in the room, environmental distractions or a difference in perceived reimbursement value. However, there is some evidence that MTurk participants can have higher levels of attentiveness and performance than laboratory participants (*Hauser & Schwarz*, *2015*). One might expect age to play a role in longer RTs, yet we saw no correlation between age and median Go RT. The fact that we saw no increase in No-Go accuracy as a result of the longer RTs, as would be expected, suggests that the difference is artificial and due to technical reasons. Although we used the same experimental platform to test both groups, there are still several potential sources of slowing such as differing operating systems, keyboards and web browsers (*Neath et al.*, *2011*; *Plant & Turner*, *2009*; *Woods et al.*, *2015*).

Despite the difference in RTs, we saw no interactions between site and task variant, and there were no unusual patterns of performance between the two groups. Our results show that online cognitive testing can produce valid and useful data, as long as one is aware of potentially longer RTs.

The online group generally rated all the task variants more highly on the subjective questionnaire. This stands in direct contrast to *Hawkins et al.* (*2013*) who reported lower engagement scores when the task was delivered online. Adjusting for age and sex eliminated the difference between the sites, implying that it was not the difference of testing location that influenced enjoyment, but rather the difference in sample composition. The theme

variant was rated particularly highly online, and this may be due to the greater levels of gaming experience in the online group.

## Comparing task variants

It is clear from our results that the theme variant was much more difficult than the other two variants, with longer RTs and lower accuracy rates. We propose several possible reasons for this: increased difficulty of spotting stimuli against the background image, a reluctance to shoot people (even cartoon characters) and the complexity of the stimuli. There was likely too much overlap in colour and pose between the civilians and cowboys, resulting in a slower categorisation of Go and No-Go stimuli.

Our motivation for using red and green objects as opposed to simple stop and go symbols was to match the intuitiveness of stop and go stimuli across task variants (i.e., we felt that shooting the cowboys and avoiding the innocents would be so intuitive that the non-game condition would need equally intuitive stimuli). However, the association between red/green and stop/go may have been stronger than we expected (see *Moller, Elliot & Maier*, *2009*) and there is evidence that attending to colour is easier than attending to shape (*McDermott, Pérez-Edgar & Fox*, *2007*). These factors may have made the points and non-game variants easier than anticipated, although any implicit association between red/stop and green/go may have been unnoticed by participants as they reported that stimuli in the theme variant were equally intuitive to those in the points and non-game variants.

The clear differences between the theme and non-game variants invalidates the use of these stimuli for gathering data comparable to non-gamelike GNG tasks. This represents an important finding since several previous studies have used complex stimuli, such as robots and monsters, in their gamified cognitive tasks (*Dörrenbächer et al.*, *2014 Dovis et al.*, *2011*; *Prins et al.*, *2011*). The idea of using graphics alone to gamify a task is not uncommon, but future researchers must ensure that the addition of gamelike stimuli does not make their task considerably more difficult. Detrimental effects on participant performance resulting from the introduction of gamelike features have been found before (*Katz et al.*, *2014*; *Miranda & Palmer*, *2014*), and it is likely that complicating a task too much may increase its difficulty such that the data it collects becomes incomparable to data from a traditional task.

*Boendermaker, Boffo & Wiers* (*2015*) investigated the use of gamelike features in a GNG alcohol-bias training task, and although they saw no overall effect of the training, they also found no evidence of a difference in training efficacy between the gamelike and non-gamelike variants. Their gamelike variant was themed and contained points, lives and levels, so their results stand in contrast to our more minimal theme variant which had a negative impact on participant performance. This inconsistency is likely because Boendermaker's task clearly delineated the stimuli from the themed surroundings of the game, i.e., using extrinsic fantasy (*Malone*, *1981*), rather than actually gamifying the stimuli as we did.

When we consider the data collected by the non-game and points variants, the Bayesian analyses we performed provide good evidence that these tasks produced equivalent data. Our points system was not particularly punishing and this may explain why we saw no impact of the points system on behaviour. There is evidence that a GNG task which rewards participants for fast responding and punishes them for failed inhibitions can optimise

performance (*Guitart-Masip et al.*, *2012*), but our study did not detect any improvement in data as a result of the points mechanic. Nevertheless, the points variant received the highest total score both online and in the lab, implying that points are a highly enjoyable game mechanic. This is interesting because adding points to cognitive tasks in order to make them more engaging is not uncommon, but to the best of our knowledge this is the first study to directly compare the appeal of points against another game mechanic.

Finally, it is clear from our results that the addition of even a single gamelike feature makes a huge difference to the participants' perception of the task. The non-game control was rated as far more boring, far less enjoyable and less mentally stimulating than either of the task variants. Although our results show this theme to be of secondary appeal to points, this may be inaccurate given that the theme variant was more difficult. As such, it comes as no surprise that participants rated it as more frustrating and felt they performed less well. Future work might investigate the role of theme more effectively by carefully controlling task difficulty. We also highlight the need for replication of our findings, with points being compared against other themes or in other contexts, such as longitudinal studies.

## Limitations and conclusions

We consider the difference in difficulty between the theme variant and the other task variants to be the most important limitation of this study. This difference is informative because gamelike stimuli and complex visual environments are common in gamified tasks, and our results highlight the need to limit the impact of these features. Although, clearly, such variations in accuracy mean we are limited in the manner in which we can compare task performance across variants. Secondly, we opted for a between-subjects design which does not allow us to study the impact of different gamelike features on an individual's performance, confounds hardware/individual differences with effects caused by the task variant. Nevertheless, the large sample size we achieved using online testing helps to counteract the lack of power associated with our experimental design. We also acknowledge that our design is not suitable to validate our task for the measurement of response inhibition, and that we would require a within-subjects design in order to test predictive validity (see also *Kato* (*2013*) and *Boendermaker, Prins & Wiers* (*2015*)). Thirdly, the task we used was quite short in duration, meaning that participants may not have had time to become bored enough to affect the data, even when playing the non-game variant. If participants were not that bored by the task, then this may have limited the potential effects of gamification. Future research might explore whether longer task durations result in greater boredom, and therefore greater impact of gamification. Finally, although we intended the questionnaire to measure enjoyment and engagement, the fact that it is delivered after the task means that the scores likely represent only a post-hoc appraisal of enjoyment of the task. In future work we intend to use a more objective measure of engagement, such as a dual task paradigm, to test for differences between the variants.

In conclusion, we found points to be a highly suitable game mechanic for gamified cognitive testing, in that they do not disrupt the validity of the data collected and yet increase participant enjoyment. This will be welcome news to experimenters who need to increase participant engagement. For example, several recent studies have used GNG tasks

to train automatic inhibition to specific stimuli, such as food or alcohol (*Jones et al.*, *2014*; *Lawrence et al.*, *2015*; *Veling et al.*, *2014*). Such studies require participants to complete several sessions of GNG training and therefore rely on high levels of engagement; our data suggest that simply adding points may achieve this goal.

Despite some hope that gamelike features might increase engagement to the point where participant performance improves, we found no evidence of such an effect in this study. We also found that while participants enjoyed the themed task and its visually interesting stimuli, the complexity of categorising such stimuli can adversely affect participant performance. Finally, we found differences in the data collected online and in the lab, with slightly longer participant RTs online, but we saw no interactions or unusual patterns. This suggests that online crowdsourcing is a very acceptable method of data collection for this type of research.

### Funding
The authors are members of the United Kingdom Centre for Tobacco and Alcohol Studies, a UKCRC Public Health Research: Centre of Excellence which receives funding from the British Heart Foundation, Cancer Research UK, Economic and Social Research Council, Medical Research Council, and the National Institute for Health Research, under the auspices of the UK Clinical Research Collaboration. This work was supported by the Medical Research Council (MC_UU_12013/6) and a PhD studentship to JL funded by the Economic and Social Research Council and Cambridge Cognition Limited. The funders had no role in study design, data collection and analysis, decision to publish, or preparation of the manuscript.

### Grant Disclosures
The following grant information was disclosed by the authors:
British Heart Foundation.
Cancer Research UK.
Economic and Social Research Council.
Medical Research Council: MC_UU_12013/6.
National Institute for Health Research.
Cambridge Cognition Limited.

### Competing Interests
The authors declare there are no competing interests.

### Author Contributions
- Jim Lumsden conceived and designed the experiments, performed the experiments, analyzed the data, wrote the paper, prepared figures and/or tables, reviewed drafts of the paper.
- Andy Skinner and Andy T. Woods contributed reagents/materials/analysis tools, wrote the paper, reviewed drafts of the paper.
- Natalia S. Lawrence and Marcus Munafò wrote the paper, reviewed drafts of the paper.

## Human Ethics

The following information was supplied relating to ethical approvals (i.e., approving body and any reference numbers):

Ethical approval was obtained from the Faculty of Science Research Ethics Committee at the University of Bristol (approval code: 22421).

## Data Availability

Bristol data Repository: 10.5523/bris.1hjvqlpbtrk961ua9ml40bauie

## Supplemental Information

Supplemental information for this article can be found online at http://dx.doi.org/10.7717/peerj.2184#supplemental-information.

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
