# Peer review of "The effects of gamelike features and test location on cognitive test performance and participant enjoyment"

_PeerJ, doi:10.7717/peerj.2184_

## Round 0.1 · original submission · Minor Revisions

The two reviewers have provided some excellent and detailed feedback. Both have mentioned a number of minor points that require clarification. These should all be straightforward to address. They also note a number of recent studies that would be worth citing in the Introduction and/or Discussion.

Reviewer 2 did suggest (Point 28) removing the Bayesian analysis entirely to make the Results section more concise. My feeling, however, is that it is worth retaining as it helps the reader distinguish between an absence of effects versus mere absence of evidence.

A few minor comments of my own:

Line 45: Missing "of"

Line 63: "participants'" not "participant's" I think

Line 71: I understand that you want comparable software across testing sites. However, it would be useful if you could cite evidence for the comparability (or otherwise) of Xperiment with other software typically used in lab-based experiments (eg Presentation, ePrime, PsychToolbox).

Line 135: Missing "the"

Line 167: Can you elaborate what you mean by a Continuous Visual Analogue Scale. I presume participants responded with a mouse click along a horizontal line. Please confirm and indicate how/whether the line was divided (eg were there markers at regular intervals along the line).

Line 261: Please clarify that these data are in Figure 3.

Line 373: Is there any evidence to suggest that non-gamers and/or females find shooting cartoon characters distasteful?

One other point that might be worth alluding to in the Discussion as a potential limitation as well as an avenue for future research: Although the basic task is intrinsically quite boring, the session is still relatively short. It may be that gamification will show greater benefits if the test is longer and/or if it is given to young children or individuals with developmental disabilities who may start 'switching off' much sooner than participants like those in the current study. My own (admittedly anecdotal) experience testing these populations is that gamification helps because it allows you to "dress up" the same experiment as lots of different shorter games that are testing essentially the same thing.

·

Basic reporting

Overall, the manuscript is well written and clear.

The raw data are supplied. However, some of the engagement questions (2,3,6, & 8) are analyzed as reverse coded in the paper. If the authors intend to make this dataset public, this coding should be included in the notes (e.g., for each item, identify that it was already reverse coded or identify that it would need to be reverse coded to make a composite score).

Figures: Overall, the figures are clear. However, one box plot is included only in the supplemental material that shows how variable the responses are, even with similar means in some conditions. This difference in the variability of the responses shown in the box plot is directly relevant to the results and interpretation. Thus, box plots should be used in the main document instead of the bar graphs for figures 2, 3, and 4 (this may not be possible for figure 5 due to the number of bars).

For Figure 5, either label the scale for the Y axis or better describe the scale used in the figure caption (e.g., if this is a percent value, then the Y axis labeling is inconsistent with figure 3).

Experimental design

Overall, the experimental methods and the research questions addressed are clear. The authors preregistered their methods, allowing for examining deviations from registration.

While most of the deviations from the preregistered methods are discussed, the pre-registered methods included one research question and analysis that is not discussed in the manuscript: “Do RTs get longer as the test goes on? Do commission errors get more frequent as the test goes on?” The authors should consider discussing why this analysis was dropped.

Validity of the findings

Overall, the results are clear. However, for the Frequentist analyses finding significant group differences (e.g., ANOVAs and post-hoc t-tests), please report the effect sizes where possible to aid with interpretation.

The analysis related to the subjective questionnaire, (Lines 302-312) found differences related to site that are described as being potentially related to background characteristics. However, the discussion (Lines 369-373) speculates beyond the analysis that was done. The authors should consider running exploratory analyses to more directly test the relation between previous game playing and enjoyment.

Additional comments

Overall, the discussion is clear. However, several relevant recent articles may aid in interpreting the results:

Related to the impact of point systems and game mechanics on performance: Attali & Arieli-Attali (2015) found that adding a point system to a math task increased speed, but not accuracy, for both adults and children. In addition, Ninaus, Pereira, Stefitz, et al., (2015) found accuracy improvements from implementation of game mechanics.

Changing the stimuli type in similar cognitive tasks has been examined in preschool children (McDermott, Perez-Edgar, & Fox, 2007) and adults (Teslovich, Freidl, Kostro, et al., 2014), finding that stimuli type had an impact on performance. For example, McDermott et al., (2007) found that preschool children are faster and more accurate on Flanker tasks when they attend to the color (red and green) rather than the shape of the stimuli (such as stars and triangles). Teslovich et al. (2014) found that participants were faster when the “go” stimuli was food (and the no-go was toys), compared to when the “go” stimuli was the toys (and the no-go was the food).

·

Basic reporting

As far as I can tell, the manuscript presented adhere to PeerJ policies. Altogether, I believe this article to be generally well written. I do have some (small) suggestions for elaboration on the background (see general comments elsewhere), mainly pertaining to inhibition, as well as some suggestions about the use of the figures.

Experimental design

The experimental design seems sound, well thought of and the research seems well executed. Although the game has several shortcomings, these are sufficiently addressed in the Discussion section. I do have several specific comments and suggestoins, as specified in the general comments section.

Validity of the findings

see General Comments.

Additional comments

I believe this manuscript to be well written and to reflect a sound research design. I do have several comments, questions and suggestions I would like to see addressed:

1. Abstract, line 19-21: please specify how performance was affected (i.e., no-go error percentages), as this is one of the more important findings.
2. I think the article would improve if a small section is dedicated to explaining what inhibition is and how exactly a Go/No-Go paradigm taps into that. This also relates to the point made in the discussion as to why it is important to ensure that the addition of gamelike elements does not render the task more difficult (as you say on line 403).
3. Related to comment 2, I would also suggest spending a few more words on the potential dangers of making changes to evidence-based paradigms, and the importance of (re)validation, as I believe this to be an essential, and often overlooked aspect of gamification. One article that might be of use there is Boendermaker, Prins, and Wiers (2015). Cognitive Bias Modification for Adolescents with Substance Use Problems – Can Serious Games Help? Journal of Behavioral Therapy and Experimental Psychiatry, 49, 13-20 (but please don’t feel obligated to use the reference only because of this comment!).
4. Have you considered looking at correlations of each Go/No-Go variant with other measures of inhibition? This might help validate the various task versions beyond the comparison in terms of reaction times and error rates.
5. I would suggest distinguishing between (gamification of) cognitive training tasks and cognitive assessment tasks more clearly, early on in the Introduction, just to prevent any misconceptions as to what the article is about. I believe gamification of these two types of cognitive tasks can be very different in terms of assessing efficacy.
6. On line 36 you reason that a shift towards participants’ goal being to try to succeed at the game would ‘thus’ result in better data. First, what exactly is ‘better data’? And second, I’m not sure that having such a goal would automatically/necessarily lead to, what I would perceive to be better data. As such, please specify here what you mean exactly and I would suggest to either explain why this would ‘thus’ lead to better data, or make this inference less strong.
7. On line 45, it seems there is a word missing: “..the effects two different..” may need an “of” in the middle.
8. On line 55-57 you mention that some studies have compared the consistency of data. As with the aforementioned ‘better data’ notion, I think a brief explanation of what is meant here with consistency would greatly improve the reader’s understanding of the matter.
9. Similarly, the examples of factors that may influence data consistency, mentioned on line 58-60, may need a bit of elaboration, e.g., how is it that differences in hardware are expected to influence performance systematically (unless of course there is a systematic difference between conditions). I think this may help the reader to understand the importance of such issues.
10. In the design section (line 81), a specific expectation is mentioned about differences in No-Go accuracy, but what about expectations about Go-accuracy?
11. Please explain why you expect the rating difference between GNG variants to be smaller in the online sample (line 84).
12. Please explain why different amounts of compensation were used for online participants ($1.50) versus those in the lab setting (£3.00), and please address how this may have affected participant’s motivation differently in these conditions.
13. Reading the Methods section, I wondered whether you also inquired about participants’ previous gaming experience, as this may influence their rating of the GNG task versions, but this is not mentioned. However, in the discussion section (lines 343 and 369), you do mention having some kind of data in that direction. Please elaborate on these measures in the Methods and Results sections.
14. In the Methods section, where the stimuli are described, it is mentioned that they are also Go or No-Go signals, based on their content (line 116), which later is revealed to be green or red objects. Please specify more clearly the link between these colour and the Go, or No-Go signal, respectively (as at this point I have to assume/infer that the greens are the Go-signals). Please also mention whether or not these object-signal mappings were counterbalanced over participants.
15. The ITI (line 117) was variable between 500-1000 ms; did you use a fixed set of values in between (e.g., 600, 700 ms, etc.) or was it purely random (e.g., 501, 622, 789 ms, etc.)?
16. On line 121 it is mentioned that there were 5 sub-blocks to each of 5 blocks. There were one-minute breaks in between the blocks, but was there any form of break in between these sub-blocks as well?
17. Please explain why no points were given for a successful inhibition (i.e., a non-response on a No-Go trial, line 138). As the task instruction is both to react as quickly as possible to Go-signals, as well as to inhibit on No-Go trials, rewarding only part of the requested behavior may be of influence. Moreover, given that there was no ‘punishment’ for making errors, participants may have inferred that the Go-trials were more important than the No-Go trials.
18. When participants quit the task before finishing, no data was recorded (line 177); however, was there at least an entry in the log that there was a drop-out occurring, or is there no way to tell how many people dropped out like this? If there is data available, please report it, as it may be an informative measure of enjoyment or frustration in each condition.
19. On line 191 you state that ‘No previous studies have investigated differences in data produced by gamelike and non-gamelike GNG tasks’. This is not the case exactly, please see e.g., Boendermaker, Boffo, and Wiers, (2015). Exploring Elements of Fun to Motivate Youth to do Cognitive Bias Modification. Games for Health Journal, 4 (6), 434-443. (Again, not intended as a must-include reference, but the statement made here may be overly strong).
20. On line 193, the sample size is discussed. Was an a priori power analysis performed (e.g., based on ‘regular’ Go-No-Go assessment studies)? Having a substantially larger sample in the online setting may also influence the chances of detecting effects.
21. Regarding the analyses, I was wondering about any ANOVA assumption violations (specifically, normality) because of the strong floor/ceiling effects in the accuracy data. If there were violations, how were these handled? Were non-parametric options considered? And finally, were there any outliers, and how were they detected and handled? Please provide this information.
22. On line 217 you mention that participant ratings were assessed visually. I was wondering to what end this was done (as ANOVA’s were also run).
23. Based on line 219, I assume only a total score of the questionnaire was analyzed. However, as these are fairly diverse questions, did you perform any form of principal component analysis on the data to see if there is more than one factor? i.e., is it certain that the questionnaire only measures one single construct?
24. I would suggest reporting the median screen resolutions instead of the means, as these are unusual numbers (line 244).
25. Please report effect sizes in the Results section wherever possible.
26. Line 256 mentions that the theme variant was ‘different’; please specify how it was different (i.e., on what measure, in which direction?)
27. Please consider adding asterisks to the figures to show significant differences.
28. Consider making the Results section a little shorter, e.g., by using fewer figures, focusing more on the significant results (mentioning the non-significant results briefly as being ns) and/or leaving out the Bayesian analyses. Although I certainly appreciate the use of Bayesian statistics, they do not seem to add much value (in terms of new, relevant information) here, as they are either in line with the ANOVA results, or had insufficient data. As such, consider leaving them out to improve conciseness.
29. An overall thought I had while reading the discussion is whether to interpret these results as ‘so, points are good’, or as ‘so, this particular game didn’t really work’. Although you address several good limitations in the Discussion section, I do wonder how you can distinguish between these two possible conclusions.
30. Regarding the (valid) point made on lines 371-373, was any of this also supported by a gender difference in the appreciation scores?
31. I disagree with the notion that the more complex stimuli in the themed variant, which led to greater variation in accuracy scores, would thus be (more) useful in assessing inhibition (line 394-396). I believe this increased variation could just as easily be due to differences in, e.g., participants’ sensitivity to distraction. As such, I would not mention this as a positive side, based solely on these results.
32. Regarding the notion on line 416-418 that this would be the first study to directly compare points to game mechanics, I would think the article by Dovis at al. (2011), referenced on line 401, who admittedly used money instead of points, would come very close. In any case, I would be very interested to see how you would relate your study to theirs.
33. Line 422: I would advise to refer to ‘this (or our) theme’, instead of themes in general, as I believe it cannot be concluded from this study alone that any theme would have the same relation (being of secondary appeal) to points.
34. Looking at Figure 5, and in the absence of effect sizes, I am a bit hesitant about the notion that points, specifically, ‘greatly increased’ participant enjoyment (line 445).
35. You could consider elaborating on the value of these results for future research and serious game designers a bit more. For example, do you have a specific recommendation?

---

## Round 0.2 · Minor Revisions

Thanks for the very detailed responses to the very detailed reviews!

There were a couple of points where you left things to "editor's discretion". Unless noted below, I'm happy with the manuscript as it stands.

There are just a couple of very minor outstanding issues:

28. The reviewer mentioned a paper by Boendermaker et al 2015. In your response you say “We will add a mention of the work to the introduction.” However I could only find a brief mention of the paper in the Discussion and it wasn't clear from your description what their study involved. I'm sympathetic to the fact that the Boendermaker paper is very new but it would still be useful to provide a little more detail in the Discussion regarding the overlap between your study and theirs - both in terms of methods and results.

34. Scanning through I noticed one effect size missing on line 328 of the clean manuscript. Please add this and double check there are no other effect sizes missing.

48. I think it would be useful to include these preregistered analyses (even if only in the supplementary information)

---

## Round 0.3 · accepted · Accept

All requested revisions have been addressed.